# Understanding Prion Strains: Evidence from Studies of the Disease Forms Affecting Humans

**DOI:** 10.3390/v11040309

**Published:** 2019-03-29

**Authors:** Marcello Rossi, Simone Baiardi, Piero Parchi

**Affiliations:** 1Department of Experimental, Diagnostic and Specialty Medicine (DIMES), University of Bologna, 40138 Bologna, Italy; marcello.rossi@ausl.bologna.it; 2IRCCS Istituto delle Scienze Neurologiche di Bologna, 40139 Bologna, Italy; 3Department of Biomedical and Neuromotor Sciences, University of Bologna, 40123 Bologna, Italy; simone.baiardi6@unibo.it

**Keywords:** prion, strain, Creutzfeldt*–*Jakob disease, human prion disease, VPSPr, GSS, fatal insomnia, experimental transmission

## Abstract

Prion diseases are a unique group of rare neurodegenerative disorders characterized by tissue deposition of heterogeneous aggregates of abnormally folded protease-resistant prion protein (PrP^Sc^), a broad spectrum of disease phenotypes and a variable efficiency of disease propagation in vivo. The dominant clinicopathological phenotypes of human prion disease include Creutzfeldt–Jakob disease, fatal insomnia, variably protease-sensitive prionopathy, and Gerstmann–Sträussler–Scheinker disease. Prion disease propagation into susceptible hosts led to the isolation and characterization of prion strains, initially operatively defined as “isolates” causing diseases with distinctive characteristics, such as the incubation period, the pattern of PrP^Sc^ distribution, and the regional severity of neuropathological changes after injection into syngeneic hosts. More recently, the structural basis of prion strains has been linked to amyloid polymorphs (i.e., variant amyloid protein conformations) and the concept extended to all protein amyloids showing polymorphic structures and some evidence of in vivo or in vitro propagation by seeding. Despite the significant advances, however, the link between amyloid structure and disease is not understood in many instances. Here we reviewed the most significant contributions of human prion disease studies to current knowledge of the molecular basis of phenotypic variability and the prion strain phenomenon and underlined the unsolved issues from the human disease perspective.

## 1. Introduction

Prion diseases, also known as transmissible spongiform encephalopathies (TSEs), belong to a group of neurodegenerative disorders of humans and animals, characterized by aggregation and tissue deposition of a misfolded form of the prion protein (PrP). In this pathological process, the physiological cellular PrP (PrP^C^) converts into abnormal PrP (PrP^Sc^) through post-translational events that lead to an increased β-sheet conformation [1]. Once formed, PrP^Sc^ replicates itself by a seeded-conversion mechanism in which PrP^Sc^ binds to PrP^C^ and mediates its conversion to PrP^Sc^ [2]. Newly converted PrP^Sc^ then propagates and accumulates preferentially, but not exclusively, in the central nervous system.

Despite their rarity, prion diseases include a wide range of clinicopathological phenotypes. They also often propagate after inoculation into susceptible hosts, a property that led to the isolation and characterization of different prion strains. These were initially defined as animal or human “isolates” that, after injection into syngeneic hosts, cause diseases with distinctive characteristics, such as the incubation period, the pattern of PrP^Sc^ distribution, and the regional severity of neuropathological changes [3]. However, following the demonstration that any given protein in the amyloid state, including those involved in other neurodegenerative diseases, can form ordered aggregates with distinct conformations, the term is frequently applied, with a much broader meaning, to all protein amyloids that show heterogeneous structures and some evidence of in vitro or in vivo propagation by seeding [4].

The characterization of distinct isoforms of PrP^Sc^ that strongly correlate with the disease phenotype also led to the concept of molecular strain typing, in which the different PrP^Sc^ isoforms or types may serve as molecular markers for prion strains [5].

The clinicopathological spectrum of human prion diseases is traditionally classified in major disease groups, which include Creutzfeldt–Jakob disease (CJD), by far the most common; fatal insomnia (FI); Gerstmann–Sträussler–Scheinker disease (GSS); and variably protease-sensitive prionopathy (VPSPr). However, apart from FI, which truly represents a distinctive disease subtype that is linked to a specific prion strain, all of the others represent heterogeneous groups including multiple distinct disease phenotypes or subtypes.

Human prion diseases also differ in their apparent origin, which is another common criterion used for their classification. Sporadic prion disease (also known as idiopathic) comprises more than 80% of all cases with an overall prevalence of about 2 cases per million and is thought to originate spontaneously from unknown stochastic cellular events leading to PrP^C^ conversion into PrP^Sc^. However, the involvement, at least in some cases, of an exogenous trigger of PrP^Sc^ formation has not been completely ruled out. The genetic form (also called inherited or familial prion disease) is linked to autosomal-dominant mutations in the prion protein gene (*PRNP*) and account for about 10% of cases. Finally, the acquired form may originate from accidental human-to-human transmission as in iatrogenic CJD (iCJD) or ritual cannibalism as in kuru, or from ingestion or inoculation of bovine spongiform encephalopathy (BSE)-derived prions as in variant CJD (vCJD) [6].

Here we have reviewed the most significant contributions of studies on human prion diseases to current knowledge of the molecular basis of phenotypic variability and the prion strain phenomenon and underlined the unsolved issues from the human disease perspective.

## 2. Historical Background

The origin of the concept of strains in the field of TSEs, a term originally intended to define the variability of an infectious agent due to nucleic acid sequence variations, dates back 60 years. At that time, the successful experimental transmission of ovine scrapie to goats, achieved by Pattison and Millson, led to the description of the “scratchy” and “nervous” (later defined as “drowsy”) phenotypes and the demonstration that they were maintained after serial passages [7]. Follow-up experiments, based on transmissions to rodents, further highlighted the heterogeneity of the scrapie agent and the phenotypic diversity of TSEs [8,9,10].

At variance with scrapie, the study of human prion strains and their characterization started much later. Early studies in non-human primates aimed to prove the transmissibility of human prion disease but did not include detailed comparative studies of the resulting phenotype in the affected animals [11,12]. The difficulties in obtaining efficient propagation in wild-type rodents because of the “species barrier” also significantly contributed to this delay [13,14]. As a consequence, the progression of knowledge concerning the basis of phenotypic heterogeneity in ovine and human prion diseases followed an opposite trajectory. Indeed, in humans, the detailed description of the main phenotypic variants of the disease preceded the characterization of prion strains after experimental transmission [15]. The development of models based on transgenic (Tg) lines allowed researchers to overcome the issue of the species barrier, and it is not surprising that the study of strain-specific properties of human prion isolates has mainly been conducted in Tg mice [16]. As the only exceptions, a few studies analyzed the transmission proprieties of human isolates in various species of non-human primates and in the bank vole (*Clethrionomys glareolus*), which only recently was proven to be more susceptible than other rodents to human prion strains [17,18,19].

Although knowledge about the phenotypic heterogeneity of human prion diseases dates back several decades, the discovery of the main molecular determinants of this variability was only achieved in the 1990s [20]. At that time, a wealth of evidence was gathered indicating that the clinical and pathological features of prion disease depend on two major molecular determinants: the *PRNP* genotype, which is specified by mutations and polymorphic sites, and the physicochemical properties (see paragraphs 3 and 4) of the PrP^Sc^ aggregates that accumulate in the affected tissues. From a historical perspective, the discovery of GSS-specific *PRNP* mutations and the demonstration that the genotype at the polymorphic codon 129 in the mutated *PRNP* allele is responsible for the strikingly different phenotypes between two familial prion diseases linked to the same mutation (i.e., D178N), first demonstrated the role that the *PRNP* genotype plays in genetic prion disease [21,22]. Similarly, the finding that the codon 129 valine (V) allele specifically correlates with a plaque-like pattern of PrP^Sc^ deposition in both sporadic and iatrogenic CJD further highlighted the central role of the polymorphic codon 129 [23,24,25]. Most significantly, the demonstration that, in both CJD and GSS [26,27,28], PrP^Sc^ molecules with distinct physicochemical properties correlate with distinct clinicopathological disease variants that are also independent of the *PRNP* genotype laid the foundation for the first robust classification of the disease and the subsequent characterization of human prion strains by transmission studies.

## 3. Molecular Basis of Phenotypic Variability and Disease Subtypes

### 3.1. Creutzfeldt–Jakob Disease (CJD) and Fatal Insomnia (FI)

In CJD and FI, the proteolytic cleavage of PrP^Sc^ aggregates by proteinase K (PK) generates two major core fragments, named types 1 and 2 (Figure 1), and a variable amount of less represented, shorter N-terminally truncated fragments.

Type 1 PrP^Sc^ primarily originates by PK cleavage at residue 82 and has a relative molecular mass of 21 kDa, while type 2 has a molecular mass of 19 kDa and is primarily cleaved at residue 97 [29,30]. Type 1 PrP^Sc^ is preferentially associated with methionine (M) at codon 129, and type 2 with V. The six possible combinations between the codon 129 genotype and PrP^Sc^ type (i.e., MM1, MV1, VV1, MM2, MV2, and VV2) specify the six sporadic CJD (sCJD) clinicopathological subtypes recognized by current disease classification with only a few exceptions [31,32,33]. These include the MM1 and MV1 cases, which have been merged in a single subtype MM(V)1 because of their similarities, and the MM2 subjects, which, on the contrary, may belong to two subtypes with distinctive histopathological features in the cerebral cortex and the thalamus, designated accordingly as MM2-cortical (MM2C) and MM2-thalamic (MM2T). Furthermore, the MV2 subtype characterized by the amyloid plaques of the kuru type has been designated as “MV2K” to distinguish it from the rare MV2C cases resembling MM2C. PrP^Sc^ types 1 and 2 also characterize the genetic and acquired forms of CJD, including vCJD, suggesting a common mechanism of PrP^Sc^ formation that is independent of the supposedly distinct etiology of the disease [29,30] (Figure 1).

After PK digestion, the Western blot profile of both types 1 and 2 PrP^Sc^ comprises additional heterogeneity, which fits well with the phenotypic subtypes described above. Most significantly, a further PrP^Sc^ fragment with an electrophoretic mobility of approximately 20 kDa, intermediate between types 1 and 2, and designated as PrP^Sc^ “i”, characterizes the sporadic and iatrogenic CJD cases showing amyloid plaques of the kuru type and carrying at least one M allele at *PRNP* codon 129 [18,31,32,34,35] (Figure 1). Additional fragments may form in human prion diseases based on the loss of the glycosylphosphatidylinositol (GPI) anchor or the availability of other proteolytic cleavage sites in the C-terminus. They include anchorless fragments of 18.5 or 17 kDa, representing the 21- or 19-kDa PrP^Sc^ core fragments without the GPI anchor and the so-called PrP-CTF12/13 that are C-terminal-truncated peptides of 12 and 13 kDa generated by PK cleavage at residues 162/167 and 154/156 [36,37,38,39]. Notably, both the presence and the relative amount of PrP-CTF13 vary significantly among CJD subtypes [39].

The PK-resistant fragments described above refer to the peptidic bone of PrP^Sc^; however, the heterogeneity of PrP^Sc^ also extends to the attached glycans. In homology with PrP^C^, the Western blot profile of PrP^Sc^ comprises a triad of fragments, which reflect the non-obligatory addition of one or two sugar chains [40]. It is well established that PrP^Sc^ molecules associated with the different subtypes of prion disease manifest distinct ratios of the three differently glycosylated forms (the so-called glycoform ratio). More specifically, while most CJD cases show a predominance of the monoglycosylated band, representing the so-called “pattern A”, vCJD and the familial prion diseases linked to the E200K, P102L, and D178N mutations display a “pattern B” that is characterized by the prevalence of the diglycosylated band [28,29,30,41,42] (Figure 1). However, this is a rather gross classification, given that significant differences in the PrP^Sc^ glycoform ratio also feature between subtypes of both groups. For example, within the first group (pattern A), the glycopattern differs significantly between sCJD MM2C and MM2T and between sCJD VV2 and MM(V)1, while it differs between genetic CJD (gCJD) D178N-129M, D178N-129V, and E200K-129M in the second (pattern B) [29,43,44].

The phenotypic spectrum of CJD includes variants showing the co-occurrence of types 1 and 2 PrP^Sc^ or even the coexistence of subtypes linked to the same PrP^Sc^ type (e.g., MV2K+2C) (Figure 2).

This scenario represents an intriguing and mostly unexplained event, which is likely relevant to the understanding of prion strains. Several studies dealt with it, and the resulting data raised some debate in recent years concerning the extent of the phenomenon. Indeed, the calculated incidence of sCJD cases with the co-occurrence of PrP^Sc^ types varied from 12% to 44% among studies that used standard antibodies against PrP C-terminus [45,46,47,48,49,50], but increased to 100% in those that used antibodies, thought to be selective for type 1, that recognize an epitope between residues 82 and 96, which is cleaved off by PK in type 2 PrP^Sc^. These findings led to the proposition that the co-occurrence of PrP^Sc^ types 1 and 2 might involve the totality of CJD patients [51,52,53]. Specifically, some researchers argued that the calculated incidence of “mixed” sCJD directly reflects the extent of brain sampling and the sensitivity of detection of a minority PrP^Sc^ type in the presence of the larger amounts of the dominant type [51]. However, we and others raised concern about the reliability of “type 1 selective” antibodies, given that, especially in mild digestion conditions, they cannot distinguish the bona fide type 1 and 2 “core” fragments from those generated by an incomplete protease digestion of the N-terminus [54,55]. By applying both an extensive regional brain sampling and an accurate and reliable methodology for the detection of the PrP^Sc^ type coexistence, which provided good sensitivity combined with absolute specificity for the bona fide type 1 and 2 “core” fragments, we confirmed an overall prevalence of types 1 and 2 co-existence of 35% in a largely unselected series of 200 cases [32,54]. Although the co-occurrence of types 1 and 2 characterizes all codon 129 genotypes, the two abnormal PrP^Sc^ isoforms coexist more frequently in MM (43%) than in MV (23%) or VV (15%) subjects (Figure 2). Moreover, the co-occurrence of subtypes linked to the same PrP^Sc^ type only affects subjects carrying MM (MM2T+2C) or MV (MV2K+2C) [32] (Figure 2). Interestingly, the uneven distribution of mixed sCJD cases among subjects carrying distinct codon 129 genotypes appears to be related to the high incidence of the MM(V)2C subtype as a co-occurring subtype (Figure 2).

It is also worth noting that the deposition of either type 1 or 2 PrP^Sc^, when concurrent, as well as the association between CJD subtypes in mixed phenotypes, are not random (e.g., certain subtype combinations have never been observed, and the relative contribution of each strain to the mixed phenotype seems to follow some rules). Furthermore, phenotypic features in “mixed” CJD cases largely reproduce those seen in the “pure” subtypes (i.e., they lack novel, specific traits) with a relative abundance depending on the most prevalent PrP^Sc^ type accumulating in the brain. Based on their relative quantitative contribution to the mixed phenotypes, CJD subtypes can be divided into three distinct groups. The first would include the VV2, MV2K, and MM2T subtypes, whose phenotypic traits invariably predominate in the mixed phenotype (“dominant” subtypes), the second the VV1 variant which always co-occurs as “not-dominant” subtype, while the third the MM(V)1 and MM(V)2C subtypes, which manifest either as dominant or “not-dominant” phenotype.

### 3.2. Gerstmann–Sträussler–Scheinker (GSS) Disease and Variably Protease-Sensitive Prionopathy (VPSPr)

While CJD and FI typically present as rapidly progressive neurological syndromes, GSS and other inherited prion amyloidosis manifest a slower clinical course and show amyloid deposition rather than spongiform change in the affected brain tissue [6]. Interestingly, this significant difference in the distinctive histopathology correlates with the PrP^Sc^ properties. Indeed, while in CJD and FI the protease digestion of full-length PrP^Sc^ generates N-terminally truncated core fragments mostly retaining the GPI anchor and the attachment to the cell membrane, in GSS PrP^Sc^ digestion generates unglycosylated anchorless fragments that form aggregates in the extracellular space [26,27].

Studies showed that the PrP^Sc^ obtained by purified amyloid preparations from GSS-affected brains mainly comprise unglycosylated 7–8 kDa PrP^Sc^ fragments with ragged N- and C-termini that are primarily composed of mutant PrP [26,27,56,57,58] (Figure 1).

Interestingly, PrP^Sc^ in VPSPr also includes a prominent internal “GSS-like” PrP^Sc^ fragment migrating at ~8 kDa, besides C-terminally truncated peptides lacking the diglycosylated form of PrP^Sc^ (Figure 1). Thus, there are significant similarities in the PrP^Sc^ physicochemical properties between VPSPr and GSS, which support the proposition that the former may represent the sporadic variant of the latter [59,60]. Furthermore, GSS brains occasionally show PrP^Sc^ fragments of higher molecular weight resembling the primary PrP^Sc^ species found in CJD and FI, such as PrP^Sc^ types 1 and 2 (Figure 1). Similarly, recent studies documented a regional variability of PrP^Sc^ properties also in VPSPr, including the presence of a fully glycosylated (i.e., comprising the diglycosylated form) CJD-like 19-kDa PrP^Sc^ fragment in some cases [61,62]. These findings not only corroborate the similarities between GSS and VPSPr, but also indicate that CJD/FI and GSS/VPSPr belong to a disease spectrum rather than being considered as separate disorders with different pathogenesis.

As the most significant example of the coexistence of GSS and CJD features, GSS patients carrying the *PRNP* P102L-129M haplotype may manifest either a rapidly progressive CJD-like phenotype with both spongiform changes and amyloid plaques or a slowly progressive “pure” GSS phenotype with amyloid plaques, but no spongiform changes. Most significantly, the former phenotype displays the co-occurrence of 21-kDa (type 1-like) and 8-kDa PrP^Sc^ fragments in the affected brain, while the latter only shows the 8-kDa PrP fragment [26,27]. It is noteworthy, however, that both vCJD and CJDMV2K show amyloid plaques in the absence of the 7–8 kDa fragments Thus, amyloid plaques in human prion disease are not invariably linked to the 7–8 kDa fragments that are ragged at both termini and lack the N-linked glycosylation, which indicates that other factors must play a significant role in determining the amyloidogenic properties of PrP^Sc^ fragments.

## 4. Characterization of Human Abnormal Prion Protein (PrP^Sc^) by Conformational and Solubility Assays

The described heterogeneity of protease-resistant PrP^Sc^ fragments associated with human prion diseases, and the demonstration that they also form in a significant amount in vivo [26,63], raised questions about the pathogenic role of these peptides, including their neurotoxic potential and involvement in the determinism of prion infectivity and strain-specific properties [39]. Concerning the latter issue, most scientists interpreted PrP^Sc^ fragment variability as indirect evidence of the structural heterogeneity of PrP^Sc^, in support of the protein conformation hypothesis of prion strains. However, the direct proof of this contention, as well as the demonstration of the level of the protein structure that is primarily involved (i.e., secondary, tertiary, or quaternary conformation), are still lacking [5,30,39,64,65,66].

Pursuing the putative conformational hypothesis, scientists have exploited other biochemical assays that indirectly explore the PrP^Sc^ structure to further investigate the basis of the variability among prion disease subtypes and strains.

Original studies on experimentally transmitted scrapie strains demonstrated that PK digestion not only generates protease-resistant core fragments of different lengths but is also variably efficient in digesting the abnormal protein.

We carried out a comprehensive study on the degree of protease resistance of human PrP^Sc^ [67]. We performed a PK titration assay on purified PrP^Sc^ from the whole spectrum of sporadic prion disease subtypes, and vCJD. Our findings demonstrated that the amount of PK required to digest 50% of PrP^Sc^ (ED50) significantly differs among CJD subtypes. Specifically, PrP^Sc^ ED50 was lowest for sCJDVV1 and VPSPr and highest for sCJD MV2K/VV2 and vCJD with sCJD MM1, MM2T, and MM2C showing intermediate values. Moreover, the ED50 positively correlated with the relative percentage of diglycosylated PrP^Sc^ in both “type 1” (VV1 and MM1) and “type 2” (MM2T, MM2C, MV2K, VV2, and vCJD) CJD groups (Table 1). By contrast, the ED50 inversely correlated with the relative amount of the 13-kDa truncated C-terminal fragment generated by PK digestion [67].

In the same study, we also searched for the so-called PK-sensitive form of PrP^Sc^ (sPrP^Sc^), which was previously defined as an isoform of PrP^Sc^ characterized by a degree of protease resistance comparable to PrP^C^. The results showed that, in sCJD, only a minority of the total purified detergent-insoluble PrP^Sc^ behaves as sPrP^Sc^. Overall, sPrP^Sc^ accounted for less than 10% of the whole PrP^Sc^ in all sCJD subtypes, with sCJDMM2C showing the highest and sCJDVV2 the lowest relative amount of sPrP^Sc^ [67]. In contrast, Safar and colleagues applied an assay they originally developed capable of discriminating conformational changes (i.e., conformation-dependent immunoassay or CDI) and found surprisingly elevated levels of sPrP^Sc^, reaching up to 80% of total PrP^Sc^ in some cases [79,80]. In another CDI study, the results also documented relatively high levels of sPrP^Sc^, although with high variability among the examined cases [81]. Interestingly, despite the different methodologies and the discrepant results, all of these studies uniformly pointed to MM2C being the sCJD subtype showing the highest relative amount of sPrP^Sc^ [67,81,82].

The correct understanding of the significant discrepancy in the amount of sPrP^Sc^ detected among studies relies on methodological aspects, the operative definition of sPrP^Sc^, and data interpretation.

It is well-established that the quantification of PrP^Sc^, which is, by definition, only partially resistant to protease digestion, is profoundly influenced by the experimental conditions in which the protease digestion is performed. In this regard, the ratio between the PK concentration and the total protein amount in the sample probably differed significantly between studies. Specifically, while Saverioni and co-workers [67] carefully adjusted the PK concentration to the amount of PrP^Sc^, in the other studies an unspecified amount of PrP^Sc^ contained in a total brain homogenate was digested with a relatively high PK concentration.

Furthermore, whereas our method separated PrP^C^ based on a physical property (solubility in sarkosyl detergent) [67], Safar and colleagues [80] enriched PrP^Sc^ in the sample by sodium phosphotungstic acid precipitation. Similarly, while we obtained the estimate of sPrP^Sc^ after the physical separation of a fully PK-sensitive PrP^Sc^ fraction forming aggregates of relatively low size, CDI measures sPrP^Sc^ as a part of the whole PrP^Sc^ without defining the actual structural difference between the two putative PrP^Sc^ species. Thus, the issue of whether the sPrP^Sc^ signal detected by CDI reflects the properties of a distinct PrP^Sc^ molecule explicitly remains unsolved.

Altogether, these considerations underline the limitations of the estimates of sPrP^Sc^, especially in the interlaboratory setting, when based on a single digestion condition and without a structural or physical definition of the putative protein isoform.

Another approach with which to search for indirect evidence of structural PrP^Sc^ variants among CJD subtypes exploited the conformational stability of PrP^Sc^ aggregates using various methods to measure the degree of conformational loss. CDI, for example, quantifies the affinity of monoclonal antibodies to conformational epitopes of PrP^Sc^. Using this approach, Kim et al. [82] measured the concentration of guanidine hydrochloride (GdnHCl) that is needed to denature 50% of PrP^Sc^ ([GdnHCl]1/2) in sCJD MM1, MM2C, and VV2 cases. The results revealed a wide range of values, even within the same sCJD subtype, suggesting that each disease variant is comprised of several distinct PrP^Sc^ conformations. They also demonstrated a significant difference in the stability of PrP^Sc^ aggregates between sCJD MM1 (the most stable) and MM2C (the least stable). Finally, they showed that PK digestion has a substantial effect on the structural stability of MM2C PrP^Sc^, given that these aggregates more easily solubilize after PK digestion [82]. The extended analysis to sCJD with mixed MM1 + 2C features confirmed that the two PrP^Sc^ types have distinct [GdnHCl]1/2 values and PrP^Sc^ structural organization at the level of both the polypeptide backbone and the quaternary packing arrangements [83].

Other groups calculated the [GdnHCl]1/2 by measuring the progressive loss of PK-resistance (conformational stability assay or CSA) or the increase in solubility of PrP^Sc^ generated by increasing concentrations of GdnHCl (conformational stability and solubility assay, or CSSA). Both methodologies confirmed the higher stability PrP^Sc^ aggregates in sCJD MM1 in comparison to MM2C [55,84]. Also, CSSA distinguished between sCJD MM2C and MM2T [85]. Given the discriminatory power of CSSA, Cali and colleagues extended their study to sCJDMM1+2C. Intriguingly, when coexisting in the sample, either naturally or in an artificial mixture, PrP^Sc^ types 1 and 2 displayed similar [GdnHCl]1/2 values.

In another study, however, we failed to reveal significant strain-specific differences in the GdnHCl denaturation curve of PrP^Sc^ aggregates monitored by CSA, in a comparative analysis of all six sCJD subtypes and vCJD [68].

In the same study [68], we also compared the thermostability of PrP^Sc^ aggregates across the CJD spectrum by applying the so-called thermo-solubilization assays (TSA) [86]. At variance with CSA, the results highlighted significant strain-specific differences allowing a gross classification of CJD subtypes in three groups that were, respectively, resistant (sCJD MM1, VV2, MV2K), sensitive (vCJD, sCJD MM2C and MM2T), and highly sensitive (sCJDVV1) to thermal solubilization (Table 1). Interestingly, the analysis of CJD brains with mixed MM1+2C phenotypes by using an antibody that selectively recognizes the PrP^Sc^ types 1 or 2, showed that when coexisting, each subtype maintains the thermal solubility of the corresponding “pure” phenotype, namely sCJD MM1 and MM2C. These findings, at variance with those of a previous study with CSA [55], support the hypothesis that in CJD brains with mixed phenotypes, subtype-specific PrP^Sc^ aggregates are spatially segregated and do not influence each other [68].

## 5. Characterization of Human PrP^Sc^ Types Conversion and Seeding Activity by In Vitro Amplification Techniques

Prion amplification techniques, such as the protein misfolding cyclic amplification assay (PMCA) and the real-time quaking-induced conversion (RT-QuIC) method, take advantage of PrP^Sc^ ability to induce a conformational change of a PrP^C^ substrate leading to the formation of ordered aggregates, mimicking the process of PrP^Sc^ replication that was thought to occur in vivo. The application of these novel techniques led, on the one hand, to the successful detection of PrP^Sc^ in tissues and fluids containing a minute amount of the abnormal protein and, on the other, to a better understanding of the molecular mechanisms of prion replication.

PMCA, which was first described in 2001 by Saborio and colleagues [87], fosters PrP^Sc^ amplification through a series of alternating sonication and rest cycles that promote the fragmentation of prion aggregates and the emergence of new nucleation sites. Depending on the protocol, the PrP^C^ substrate is provided by normal brain or peripheral tissue homogenates [88,89,90], plasma [91], human platelets [92], cultured cells [93], or recombinant PrP (rPrP) [94].

The significant heterogeneity in the source of PrP^C^ substrates and experimental protocols utilized to date prevents the comparison of current PMCA studies on human prion diseases. Nevertheless, as the most significant finding, most studies revealed a higher seeding activity and amplification efficiency of vCJD and sCJD prions linked to PrP^Sc^ type 2 than in those to PrP^Sc^ type 1 [88,89,90,91,92,95,96,97,98,99,100,101,102,103]. Accordingly, only a few studies have, to date, accomplished an efficient PMCA-amplification of PrP^Sc^ type 1 [88,96,97,101,103].

By exploiting the ability of PMCA to replicate PrP^Sc^ type 2 and using specific brain regions as substrates to mimic the regional tropism of prions, Privat and colleagues [102] found that vCJD and sCJDVV2 seeds partially reproduce the specific brain targeting observed in affected brains. Specifically, they found a highly significant relationship between in vivo and in vitro regional brain targeting, suggesting an influence of both PrP^C^ level and locally expressed cofactors in strain tropism [102]. Hence, the study confirmed the utility of PMCA in exploring potential cofactors and general mechanisms of strain replication [104,105,106,107,108]. Haldiman and coworkers also employed the technique for the characterization of sCJDMM1+2C while aiming to investigate the impact of strain coexistence in PrP^Sc^ replication and adaptation. Interestingly, by applying consecutive rounds of PMCA, thereby mimicking an evolutionary process, they found that the PrP^Sc^ type 1 conformations, presenting lower stability as assessed by CDI, were preferentially amplified in comparison of that of the more stable PrP^Sc^ type 2 [96].

While the studies above confirm that PMCA may have a role in the investigation of PrP^Sc^ conversion and the molecular events leading to the acquisition of strain-specific features, its application for diagnostic purposes is limited. Indeed, apart from a test for the detection of vCJD prions in human urine and blood [90,98], the implementation of the PMCA technique to date failed to provide a robust and reproducible assay to be applied in sCJD diagnostics.

QuIC is another in vitro conversion assay that exclusively uses recombinant PrP (rPrP) as a substrate and in which sonication is replaced by shaking at a relatively high temperature. RT-QuIC, in particular, is the last adaptation of QuIC in which the seeded conversion of rPrP into aggregates of PrP^Sc^ is monitored in real time. The monitoring is permitted by including Thioflavin T (ThT) in the reaction, which binds the aggregated PrP^Sc^ causing a change in the ThT emission spectrum. RT-QuIC has ultimately gained more interest than PMCA as a detection technique with diagnostic applicability as it is less affected by the potential drawbacks of PMCA, such as the time taken, the complexity of the substrate, and the reliance on sonication, which is difficult to standardize [109]. Accordingly, the application of RT-QuIC successfully provided a robust and reproducible assay with high sensitivity and specificity for the clinical diagnosis of CJD.

Diagnostic RT-QuIC for CJD uses cerebrospinal fluid (CSF) or olfactory mucosa as seeds [110,111,112] and various sources of rPrP as a substrate, the latter being one of the main factors affecting the diagnostic performance of RT-QuIC. As the most significant example in humans, the use of truncated (90–231) hamster rPrP instead of full-length (23–231) hamster rPrP, in addition to other adjusted parameters (i.e. SDS concentration and temperature), in the so-called second generation RT-QuIC, significantly improved the diagnostic sensitivity of the assay without affecting its specificity [113,114]. The results obtained in several laboratories with the first generation of this assay demonstrated a 79–86% sensitivity and a 99–100% specificity [109,115,116,117], while the diagnostic sensitivity with the RT-QuIC of the second generation increased to 95–97% [114,118,119,120].

Recently, RT-QuIC in humans also detected PrP seeding activity in the skin, peripheral nerve, and eye of CJD patients [121,122,123].

Current results concerning the ability of RT-QuIC to discriminate among prion disease subtypes are limited. The three major sCJD subtypes, namely MM(V)1, VV2, and MV2K, demonstrated overall comparable test sensitivity and kinetics of the reactions (i.e., lag phase and ThT maximum fluorescence) [114,115,116,119,124]. In contrast, CSF seeds from sCJD MM2C, MM2T, and VV1 subtypes, as well as VPSPr, showed a higher percentage of negative responses, a lower fluorescence intensity and, when positive, an extended lag phase in comparison to the most common subtypes [114,115,116,119]. Interestingly, the reduced ability to replicate in vitro parallels the reduced capacity to propagate in vivo by experimental transmission.

Intriguingly, at variance with the amplification efficiency of PMCA, RT-QuIC showed a relatively low seeding activity of vCJD prions [114,116], which can be partially overcome by using bank vole rPrP [122,125].

Finally, several studies suggested that RT-QuIC may also be useful as a diagnostic test for iCJD and at least some gCJD variants [103,114,115,116]. Interestingly, gCJD cases carrying the common and highly penetrant E200K-129M haplotype [126] showed the shortest lag phase among the tested human prion samples [114,115].

In conclusion, the results obtained by these in vitro assays further highlight the complexity of the mechanisms regulating prion conversion and replication, which are thought to transmit structural information by a template model. Indeed, this likely involves PrP^C^ structural features and molecular factors that are unrelated to PrP^C^, in addition to PrP^Sc^ properties.

## 6. Characterization of Human Prion Strains by Experimental Transmission

Prion strains are operationally defined as infectious isolates that, when transmitted to syngenic hosts, under fixed and controlled conditions, exhibit distinct prion disease phenotypes [127]. The strain-associated phenotypic traits not only include biological and histopathological features (i.e., incubation times, histopathological lesion profiles, and specific neuronal target areas), but also biochemical properties related to the PrP^Sc^ structure (i.e., the pattern of electrophoretic mobility, glycosylation, and PK resistance), which, although not invariably, usually persist upon serial transmission [5,128].

To date, the experimental transmission of human prion disease “isolates” has led to the identification of five different CJD strains named M1, V2, M2C, M2T, and V1 according to the combination of the *PRNP* codon 129 genotype that is most susceptible to PrP^C^ conversion and the PrP^Sc^ type (1 or 2) accumulating in the brain. The same, or highly similar, strains characterize gCJD, kuru, and iCJD. Initial evidence suggests that additional prion strains may originate from GSS and VPSPr transmissions, although further studies are needed to verify this hypothesis (Figure 3).

### 6.1. Strain M1

The MM(V)1 subtype, either pure or mixed with MM(V)2C, often referred to as “typical or classic CJD”, comprises about 65% of all sCJD cases [31,32]. The first successful transmission of sCJDMM(V)1 must be reasonably attributed to Gajdusek’s group because the brain homogenate used to inoculate the first chimpanzee affected by the transmitted disease belonged to a case of sCJD later classified as such [12,18].

Prusiner’s lab first characterized the phenotype of experimentally transmitted sCJDMM(V)1 in Tg mouse lines overexpressing a chimeric mouse-human prion protein (MHu2M). The MM1 inoculum transmitted the disease with a mean incubation time of about 190 days, which decreased by 50% in MHu2M mice with amino acid substitution at residue 165 and 167 (Tg (MHu2M,M165V,E167Q)) [129]. The comparison of histopathological and biochemical PrP^Sc^ features in these mice revealed a consistent pattern of PrP^Sc^ deposition, distribution of vacuolation, and electrophoretic mobility of a PrP^Sc^ unglycosylated fragment (21 kDa, type 1) compatible with a single strain.

In full agreement with the disease epidemiology in humans, the sCJDMM1 inoculum showed a faster transmission in mice expressing human PrP transgene encoding M at codon 129 (Tg(HuPrP,M129)) than in the homologous mouse line with V at the same codon (Tg(HuPrP,V129)) [129], identifying codon 129 MM as the most susceptible genotype to this strain.

Other groups also successfully transmitted sCJDMM(V)1 to transgenic and knock-in mouse lines expressing human PrP^C^, obtaining results compatible with a single prion strain, designated as M1 [71,72,73,78,130,131] (Figure 3).

As expected, the injection of sCJDMM1(V)1 in lines of Tg mice with the three different genotypic combination at codon 129 (HuMM, HuMV, and HuVV) and expressing normal levels of PrP^C^ within the brain resulted in a more efficient transmission in Tg HuMM [71], confirming the previous results in Tg mice overexpressing PrP^C^ [129].

The review of archival material from the NIH series of transmitted disease, which proved the transmissibility of human prion disease [132,133,134,135], has also allowed the study of the transmission properties of sCJDMM(V)1 in four species of non-human primates, namely squirrel, capuchin, spider, and African green monkeys [18]. The characterization of these primary transmissions revealed strikingly homogenous results regarding the incubation time, type of spongiform changes, lesion profile, and PrP^Sc^ properties among more than 30 sCJDMM(V)1 inocula, yielding the most robust support to the contention that sCJDMM(V)1 is linked to a single prion strain.

Studies of the transmission properties of sCJDMM(V)1 in bank voles and transgenic mice expressing bank vole PrP (BvPrP) with either M or isoleucine at the polymorphic codon 109 (BvPrP-109M and BvPrP109I) also showed a highly consistent lesion profile, a pattern of PrP^Sc^ deposits and Western blot profile of PK-resistant PrP^Sc^ with substantial similarities with those observed in the human phenotype.

A rare atypical sCJD phenotype that is linked to MM at codon 129 and characterized by PrP-amyloid plaques in subcortical white matter, but otherwise very similar to the MM(V)1 subtype [136,137], has also been transmitted to bank voles. The results suggest that the M1 strain is also responsible for this variant, despite the unusual pathological features [137] (Figure 3).

The inoculation of gCJD E200K-129M brain homogenates into Tg(HuPrP,M129) and Tg(MHu2M,M165V,E167Q) showed an incubation time and a size of PrP^Sc^ fragments (type 1) comparable to those obtained with sCJDMM1 [129]. Similarly, the results obtained in bank voles inoculated with sCJDMM(V)1 and gCJD linked to E200K-129M and V210I-129M supported the view that the MM(V)1 subtype is related to the M1 strain in all forms of CJD, irrespective of their different etiology (Figure 3). The latter contention is also supported by the finding that squirrel monkeys inoculated with sCJDMM(V)1, gCJD E200K-MM1, and iCJD MM1 showed identical transmission properties [18]. Finally, the inoculation of gCJD linked to a six-octarepeat insertion in cis with M at codon 129 and PrP^Sc^ type 1 in 129VV Tg152 mice, a murine model lacking any transmission barrier to sporadic/iatrogenic forms of CJD regardless of the codon 129 genotype of the inoculum [138], also gave results closely resembling that of the sCJDMM(V)1 isolates with a 100% attack rate [139] (Figure 3).

Taken together, the results from transmission studies indicate that the same M1 prion strain is responsible for all cases of CJD MM(V)1, independent of the supposedly different etiology (i.e., sporadic, genetic, or iatrogenic).

### 6.2. Strain V2

At variance with all other CJD subtypes, the experimental transmissions of sCJD VV2 and MV2K revealed a convergence to a single propagating phenotype, indicating that a common strain, designated as V2, is linked to both subtypes [18,71] (Figure 3). Consequently, the host codon 129 genotype is the main one responsible for the phenotypic differences between these subtypes in humans. The transmission of the VV2 and MV2K subtypes to Tg mice and non-human primates carrying M at codon 129 showed a longer incubation time and a lower attack rate in comparison to those inoculated with sCJDMM(V)1, indicating a lower virulence [18,129]. However, transmission of sCJD MM(V)1, VV2, and MV2K in Tg(HuPrP,V129) mice revealed the opposite behavior [129]. Likewise, sCJD VV2 and MV2K showed the fastest transmission and the highest attack rate in Tg mice carrying HuVV [71]. Notably, the latter represents, to date, the genotype/agent combination with the shortest incubation time in mice expressing physiological PrP levels, within the spectrum of human prions (Table 1). Interestingly, the demonstration that MV2K transmits more efficiently in Tg HuVV than in Tg HuMV indicates that, regarding prion replication efficiency, the strain tropism for the V allele is more critical than the genotypic homology between the inoculum and the host [71]. 

Recent observations on iCJD in humans carrying MM, as well as the characterization of the transmission properties of the full spectrum of iCJD cases, have further highlighted the ability of the V2 strain to adapt its properties to host-expressing M at codon 129 [34,140]. At variance with sCJD, iCJD patients carrying MM at codon 129 may display two distinct phenotypes. While the most prevalent “nonplaque type” reproduces the typical sCJDMM(V)1 and is linked to PrP^Sc^ type 1, the “plaque type” shows a widespread occurrence of kuru plaques and plaque-like deposits, and is associated with a PrP^Sc^ type “i” (hence the name iCJD MMiK). Transmission studies in knock-in mice expressing human PrP with 129 MM and VV showed that nonplaque-type, dura mater graft-associated iCJD transmitted most efficiently in the former, while iCJD MMiK in the latter [141]. Moreover, in iCJD MMiK transmissions, PrP^Sc^ type “i” reproduced itself in 129MM mice, whereas it shifted to PrP^Sc^ type 2 in the 129VV mice. This observation raised the possibility that the V2 strain causes iCJD MMiK, a hypothesis that was later confirmed through the demonstration that the inoculation of sCJDVV2 in 129MM mice results in an identical phenotype, which is still reversible by reinoculation in 129VV mice [72,141]. This phenomenon, designated as traceback, elegantly demonstrated that the plaque-type iCJD depends on the propagation of V2 prions in 129MM subjects. In line with this observation, the “intermediate” pattern of electrophoretic mobility of PK-resistant PrP^Sc^ corresponding to a 20-kDa relative molecular mass of the fragment, also characterized sCJDMV2K, thus represented the hallmark of the conversion of the codon 129M allele by V2 prions.

Notably, kuru transmitted to non-human primates [18] revealed transmission properties equivalent to those of the sCJDVV2 inocula (Figure 3). Moreover, a single case of kuru from a 129MM subject showing the intermediate PrP^Sc^ profile revealed PrP^Sc^ type 2 and neuropathological features matching those seen after transmission of kuru brains carrying 129VV [142], paralleling the observations in iCJD MMiK. Altogether, these data strongly support the hypothesis that kuru originated from the consumption of brain material of an individual with sCJD VV2 or MV2K and the subsequent serial transmission of V2 prions [18].

Finally, a single gCJD E200K-129V inoculum showing PrP^Sc^ type 2 has been transmitted to Tg152, expressing human PrP 129V with a 100% attack rate, short incubation period, and plaque-like focal deposits [143], thus paralleling the results obtained with sCJD VV2 and MV2K inocula. Notably, when inoculated in Tg23 and Tg49 mice expressing human PrP with E200K-129M at 3x and 2x the PrP^C^ levels, the same case showed a marked increase of the incubation time, reduced attack rate, and less abundant PrP plaque during a neuropathological evaluation [143]. Overall, these results demonstrate that, as for the M1 isolate, the V2 strain is responsible for sporadic, genetic, and acquired CJD cases, highlighting, once again, the fact that the strain-specific properties of CJD prions are mainly encoded by PrP^Sc^ properties that are independent of the presence of specific *PRNP* mutations and the disease etiology.

### 6.3. Other CJD Strains: M2T, M2C and V1

FFI, the genetic form of FI linked to the D178N-129M *PRNP* haplotype, has a higher incidence than the sporadic form (sFI or MM2T). Accordingly, FFI was successfully transmitted earlier than sFI. FFI induced disease in both wild-type [144] and Tg mice overexpressing human PrP [64,145], whereas sFI into Tg mice expressing a chimeric mouse-human PrP [146]. Neuropathological findings and PrP^Sc^ properties in the affected mice showed striking similarities between FFI and MM2T inocula, which is consistent with a common strain (M2T) [146] (Figure 3). Data from subsequent transmissions in murine lines either overexpressing human PrP, such as Tg(MHu2M,M165V,E167Q), Tg(MHu2M), and Tg(HuPrP,M129) [129], or expressing normal PrP levels (HuMM) [74] further corroborated the conclusion.

Among the sCJD subtypes, the MM2C (strain M2C) is the only one that did not successfully transmit to knock-in mice expressing human PrP at physiological levels [71,73,74]. However, the disease propagated successfully in transgenic and knock-in murine lines overexpressing human PrP with 129M by 2 to 6 fold [78,129,130,147], although with a remarkably longer incubation time than M1 prions [17,78,129,130]. The MM2C subtype also transmitted to Tg and knock-in mice carrying the BvPrP-109M transgene [19,148], but not to Tg MHu2M [129]. Like M1 prions, M2C prions transmitted more efficiently in Tg mice expressing 129M than in those carrying 129V [129]. Taken together, these transmission data indicate that the M2C strain has a low transmission efficiency and a reduced infectious potential. In this regard, it is noteworthy that, at variance with an M1 prion, there is no demonstration to date of the human-to-human transmission of M2C prions. Moreover, whether M2C prions are also linked to gCJD, as is the case for M1, V2, and M2T prions, remains to be seen.

No study specifically addressed the transmission proprieties of VV1, the last of the successfully transmitted sCJD subtypes [71]. sCJDVV1 inoculated into Tg HuMM, HuMV, and HuVV mice expressing normal PrP^C^ levels transmitted the disease most efficiently in Tg HuVV than in Tg HuMV, whereas it did not propagate in Tg HuMM [71]. Thus, like V2 prions, V1 prions preferentially transmit to a host expressing at least one 129V allele. However, V1 prions showed significantly longer incubation times and a lower attack rate than V2 prions, which indicate a lower virulence of the former. More recently, VV1 transmitted successfully to Tg mice overexpressing human PrP^C^ levels and carrying either 129V [78] or 129M [130].

### 6.4. Variant CJD

Shortly after its discovery in 1996, two groups separately transmitted vCJD to wild-type and Tg mice [138,149], leading to the identification of a specific pathological and molecular “signature” [14], and the demonstration that a communal agent strain is responsible for both BSE and vCJD [138,149]. The transmission of vCJD to nonhuman primates [150], as well as the description of the traceback phenomenon after retro-transmission to bovinized transgenic mice (TgBov) [75], further confirmed the origin of human vCJD from BSE prions. Remarkably, while BSE transmitted efficiently to TgBov mice, a substantial species barrier existed between cattle and humans as demonstrated by the resistance of knock-in *PRNP-*humanized mice expressing normal levels of the PrP to BSE prions [151]. This species barrier only partially disappeared after secondary transmission of vCJD to these mice, indicating that the human-to-human transmission of BSE prions is relatively efficient [151,152]. However, the increased transmission efficiency was lower than expected given that the majority of vCJD infected Tg mice remained asymptomatic, despite prion replication indicated by PrP^Sc^ accumulation, a finding which was confirmed in different models [153,154]. Moreover, human vCJD prions maintained efficient transmission to Tg mice expressing ancestral bovine PrP^C^, without an obvious transmission barrier [155]. This particular behavior of the human-adapted BSE strain has been interpreted as an example of non-adaptive prion amplification [156], in which the causative ancestral BSE prions are not optimally adapted for full pathogenic potential in humans. The incomplete adaptation would possibly occur because dominant PrP^Sc^ conformers bypass the need for the selection of optimized prion conformations from larger ensembles, as it would occur for most prion strains during interspecies transmission [157].

To date, all but one vCJD patients carried MM at codon 129 [158,159]. Thus, as for the CJD strains M1, M2C, and M2T, the susceptibility for the vCJD strain is highest in codon MM carriers, while V confers protection to this strain.

vCJD transmission to humanized knock-in mice lines expressing different genotypes at codon 129 (MM, MV, VV) revealed the critical role played by codon 129 in determining the host susceptibility to the disease (i.e., the attack rate and incubation time), being highest in subjects carrying MM and lowest in those carrying VV [75,151]. Nevertheless, the V allele did not confer full protection from the disease, even in the homozygous state. The amount of PrP accumulation and of florid plaques also varied according to codon 129 genotype [75], although the PrP^Sc^ Western blot profile did not [75,151].

Transmission experiments with vCJD prions led to other significant findings, including the discovery of substantial infectivity in extraneural tissues, including tonsils, spleen, rectum, and blood components [160,161,162,163,164], and the ability of this prion strain to sustain a chronic asymptomatic infection in lymphoreticular tissue [164].

Given the causal link with the oral exposure to food contaminated with BSE-affected cattle, the experimental transmission of vCJD prions was also carried out through peripheral routes. In non-human primates challenged with brain homogenates from first-passage animals with BSE, the intravenous route showed a transmission efficiency similar to the intracerebral inoculation, but significantly higher than the oral exposure [165]. Moreover, the results documented substantial variability in the incubation periods after oral exposure ranging from 3.7 to 10 years. Intriguingly, the cases with a longer incubation time remained asymptomatic and only accumulated an atypical isoform of PrP in the lumbar spinal cord [166]. At variance with the intracerebral route, the intraperitoneal inoculation of vCJD brain tissue in Tg mice overexpressing PrP (129M) resulted in an asymptomatic infection involving peripheral tissues [164]. Interestingly, the peripheral challenge in these animals led to an early PrP^Sc^ accumulation in the spleen, followed by a plateau, while PrP^Sc^ became detectable in the brain only late in the course of the disease [164].

An intriguing result of vCJD transmission experiments concerns the claimed emergence of different phenotypes or prion “signatures” in Tg mice, largely overlapping those observed with the sCJD isolates. The phenomenon seems to mainly concern heterotypic transmissions (i.e., discordant PrP sequence between the inoculum and host) to Tg mice expressing 129V [138,153,154,167,168], although this is not the rule [164]. These results should be treated with caution as they might depend on the choice of the construct and genetic backgrounds, as well as on the expression level of the transgene. In support of this view is the finding of a single phenotype and a conserved PrP^Sc^ signature after serial transmissions of vCJD prions in wild-type mice [160]. More recently, Comoy et al. reported the emergence of a novel “atypical” phenotype mainly involving the spinal cord in rodents and nonhuman primates challenged with vCJD-infected blood [169]. The authors hypothesized a dissociation between infectivity and toxicity and suggested that the leading cause promoting the phenotypic switch might be related to non-protease-resistant and soluble PrP^Sc^ in transfused blood. Evidence of a similar dissociation after primary transmission of BSE to mice (i.e., a 100% attack rate, but PrP^Sc^ was not detectable in half of the cases) [150] and the occurrence of an analogous “spinal” phenotype associated with atypical PrP features in few macaques orally exposed to BSE, support the speculation [166].

### 6.5. Gerstmann–Straüssler–Scheinker (GSS) Disease

At variance with CJD and FI, the early attempts to transmit GSS experimentally had much less success. Indeed, GSS P105L, A117V, and the majority of P102L cases failed to propagate the disease in both rodents and nonhuman primates [134,170,171]. In the late 1990s, the generation of Tg mice with an amino acid alteration (101L) into murine PrP, equivalent to P102L in the human PrP gene, allowed the significant increase of the transmission efficiency of GSS P102L [172]. Nevertheless, GSS P102L cases lacking spongiform change failed to induce clinical disease in these mice, too, although the inoculation elicited PrP deposition in both primary and secondary passages [173]. These observations led to the hypothesis that GSS brains lacking PrP^Sc^ 27-30 and spongiform change transmit non-infectious prions, which promote PrP-amyloidogenesis, but no spongiform change [174]. Notably, at the second passage, the 101L-passaged GSS P102L prions transmitted to wild-type mice [172]. The latter finding has questioned results obtained from similar experimental transmissions of GSS P102L prions in humanized Tg overexpressing human PrP with the P102L mutation. Indeed, in these mice, GSS P102L was efficiently transmitted at the first passage (homotypic transmission) [143,175], but the secondary propagation to wild-type mice failed [175]. The results suggest that 101L-passaged GSS P102L prions [172,174] may represent a novel strain generated by the mutant mouse PrP. The finding also highlighted the need for homotypic hosts for genetic prion disorders to obtain accurate and reproducible transmissions and avoid the emergence of novel strains.

Using a similar experimental design, GSS A117V was faithfully transmitted to Tg mice overexpressing human A117V PrP; although after an extremely long incubation period, inducing both clinical disease and spongiform change. Interestingly, immunoblot analysis revealed the coexistence of PrP^Sc^ 27–30 kDa and 7-8 kDa fragments [176]. Recently, GSS G131V intracerebrally inoculated into Tg mice expressing human PrP at a level 8–16-fold higher than physiological levels induced PrP^Sc^ deposition within the brain detectable at immunohistochemistry; nevertheless, immunoblotting and RT-QuIC failed to reveal abnormal PrP [177].

GSS F198S, A117V, and P102L (the latter including both the phenotype associated with PrP^Sc^ type 1-like and that accumulating ~8 kDa fragments) have also successfully transmitted to bank voles expressing isoleucine at codon 109 (Bv109I) [178]. Interestingly, a GSS P102L case with both type 1-like and 8 kDa PrP^Sc^ in the frontal cortex transmitted two distinct phenotypes in bank voles: the first characterized by short disease duration and accumulation of 8 kDa PrP^Sc^, and the second by long duration and 21 kDa PrP^Sc^. This result suggested that GSS P102L might transmit different prion strains [178], although this speculation has not been confirmed by serial transmission experiments. 

Overall, this evidence raised the question of whether GSS is a genuinely transmissible spongiform encephalopathy and, if this is the case, whether single or multiple prion strains sustain the disease. The available data indicate that various both infectious and non-infectious prion agents with distinctive features may be responsible for the GSS phenotype. The former may induce clinical disease and spongiform change in the host brain and accumulate PrP^Sc^ with a molecular mass that is similar to CJD and FI (and, therefore, can be classified as genuine TSE), while the latter does not transmit the clinical disease, but induces PrP-amyloid deposition formed by PrP^Sc^ species of lower molecular mass (7–8 kDa).

### 6.6. Other Inherited Prion Amyloidosis

The Q226X variant is the only PrP-truncating mutation that successfully transmitted the disease in Tg mice overexpressing human PrP. The detection of abnormal PrP in the brain by immunostaining, immunoblot, and PrP amyloid seeding activity assayed by RT-QuIC faithfully demonstrated the disease transmission. Under the same experimental conditions, the Q227X variant failed to propagate the disease [177]. Similar attempts to transmit the Q163X and Y145X stop-codon variants in humanized Tg mice and wild-type rodents failed as well [170,179].

### 6.7. Variably Protease-Sensitive Prionopathy (VPSPr)

Two recent studies accomplished the transmission of VPSPr in humanized Tg mice expressing either physiological [76] or 2–8-fold PrP levels [78]. Brain inocula representative of all 129 genotypic combinations [78] or from subjects carrying the VV or MV genotype [77] yielded comparable results despite the slightly different study design. While no mice developed the clinical disease, about half of them showed PrP^Sc^ accumulation within the brain at immunohistochemistry assessment, and only one third showed detectable abnormal PrP by Western blot analysis [76,78]. Finally, there was no evidence of transmission after a second passage. These results showed striking similarities to those obtained in a subgroup of GSS cases. More recently, VPSPr inocula efficiently transmitted the disease to bank voles (Bv109M, Bv109I) [180]. Despite the low attack rate after the first passage (5% for Bv109M, 35% Bv109I), all inoculated animals developed the disease and showed dramatically reduced survival times (about 50%) after the secondary transmission, suggesting a consistent species barrier between humans and bank voles. Intriguingly, the affected animals showed three histo-molecular phenotypes, apparently unrelated to genotype at codon 129, with transmission characteristics reproducing those of PrP^Sc^ type 1 (21 kDa) and 2 (19 kDa) sCJD, and GSS expressing the ~7 kDa PrP^Sc^ fragment [180].

Further studies should elucidate whether these three phenotypes are related to different VPSPr isolates, which in humans converge in a single phenotype, or, instead, reflect the heterotypic background of the bank vole that sustains limited PrP^Sc^ structural variations independent of the predominant conformer of the human inoculum. Whatever the case, and despite the not univocal results, these transmission studies again highlighted the similarities between GSS and VPSPr. Indeed, the two diseases not only show similar PrP^Sc^ profiles including internal PrP fragments of low molecular mass and the tendency to form extracellular amyloid aggregates, but they also share the relatively limited transmission efficiency and infectious potential.

## 7. Summary and Gaps in Our Understanding of Human Prion Strains

The terms “amyloid” and “strain” are increasingly used in the field of neurodegenerative diseases to define shared pathogenic mechanisms between these disorders. Protein scientists now define amyloid as including all protein fibrils that display the cross-β fiber diffraction pattern when examined with X-rays, not only those that bind the dye Congo red and reveal a green birefringence when viewed under polarized light. Most importantly, prion or prion-like strains appear to be a general phenomenon related to amyloid polymorph, namely defined as phenotypic variants encoded by the same protein sequence, which assumes variant protein conformations. Substantial evidence also supports the idea that a prion-like, seeded-polymerization mechanism is responsible for the formation and propagation of protein aggregates in all neurodegenerative disorders [181].

Despite the advances, however, we lack a firm comprehension of the link between amyloid structure and disease in many instances. The full understanding of prion (prion protein) strains, in particular, is challenged by the complexity of human disease, which includes forms with distinct etiology and a broad spectrum of phenotypic variants (including disorders such as GSS that largely overlap with Alzheimer’s disease and other brain amyloidoses). What is the origin of sporadic prion disease? If it is caused by stochastic events related to cell aging, why do some rare CJD subtypes, such as VV1 and MM2T, mostly affect young adults? If the majority of human prion diseases originate spontaneously from random events, why is the anatomical spread of a given disease variant so consistent from case to case? What is the meaning of the ascending course from subcortical areas to the cerebral cortex that is so evident and consistent in some variants (e.g., VV2 and MM2T)? Furthermore, what is the basis of PrP^Sc^ types’ co-occurrence in the same brain? Why may the M2C strain coexist with all sCJD variants affecting the M allele and even dominate the M1 in some cases despite being the sCJD subtype that shows the lowest seeding activity? Despite many efforts, uncertainties also remain about the structural properties of amyloid aggregates that correlate with the strain-specific properties (i.e., the incubation time and rate of spread, regional targeting, lesion profile, and distinctive histopathology). What is the structural basis that explains the proteolytic generation of PrP^Sc^ type 1, type 2, type “i” and the internal fragments lacking the GPI anchor? What is the role of PrP glycans in determining the seeding and targeting properties of strains? Given the importance of the primary protein sequence in determining the amyloid structure, why is residue 129 so critical, while the majority of mutations linked to genetic CJD do not generate strain-specific properties?

Five decades have passed since Gajdusek’s pioneering studies demonstrated the transmissibility of human spongiform encephalopathies and almost four since Prusiner’s discovery of the prion protein and his formulation of the protein-only hypothesis. We hope that with the powerful tools of structural, molecular, and cellular biology available today, scientists will soon be able to answer many of these and other remaining crucial questions.

## Figures and Tables

**Figure 1 viruses-11-00309-f001:**
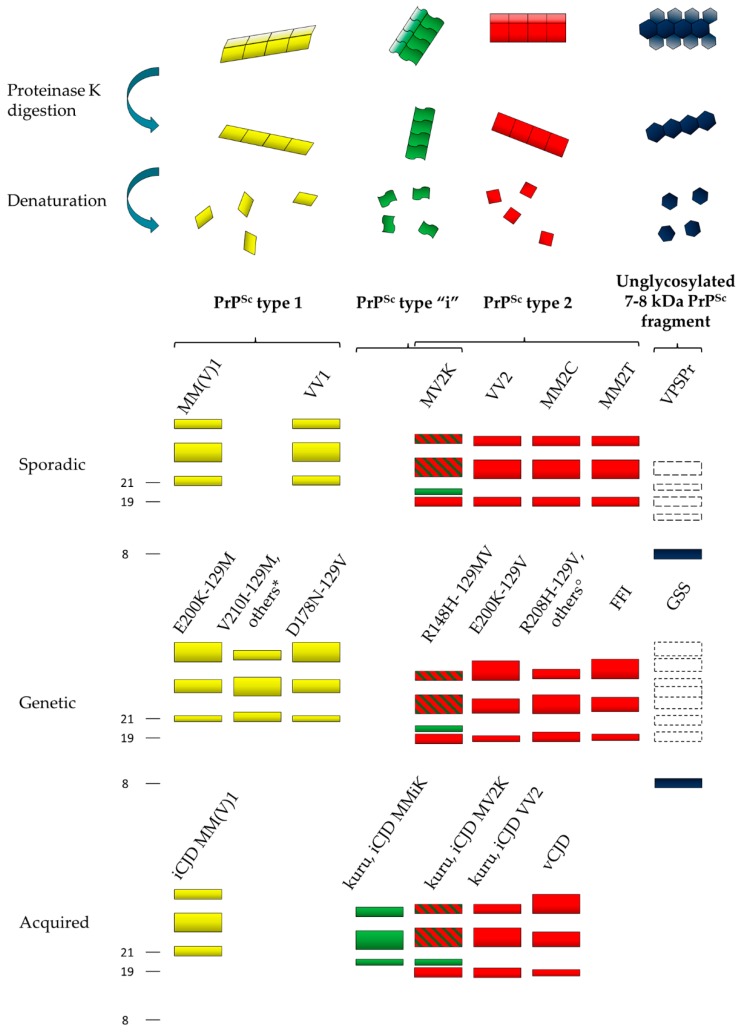
Schematic representation of the heterogeneity of abnormal prion protein (PrP^Sc^) aggregates and proteinase K (PK)-resistant core fragments in human prion disease. Four dominant human PrP^Sc^ fragments can be distinguished by sodium dodecyl sulfate (SDS) electrophoresis and immunoblotting after PK digestion and protein denaturation. They include PrP^Sc^ type 1 (21 kDa, in yellow), PrP^Sc^ type “i” (20 kDa, in green), PrP^Sc^ type 2 (19 kDa, in red) and the 7–8 kDa unglycosylated PrP^Sc^ fragments (in dark blue). The bands with dotted lines symbolize PrP^Sc^ fragments that are only occasionally seen. PrP^Sc^ types 1, 2 and “i” comprise a triad of fragments representing, from top to bottom, the diglycosylated, monoglycosylated, and unglycosylated isoforms of the protein. The predominance of the monoglycosylated or diglycosylated band defines “pattern A” or “pattern B”, respectively. The main disease subtypes associated with the four PrP^Sc^ profiles in each etiological disease group (i.e., sporadic, genetic or acquired) are listed. Other mutations include: * PrP^Sc^ type 1: G114V-129M, R148H-129M, D178N-129V, T188K-129M, V189I-129M, E196K-129M, E200K-129M, V203I-129M, R208H-129M, V210I-129M, 2-6 OPRIs–129M; ° PrP^Sc^ type 2 196K-129V, E200G-129V.

**Figure 2 viruses-11-00309-f002:**
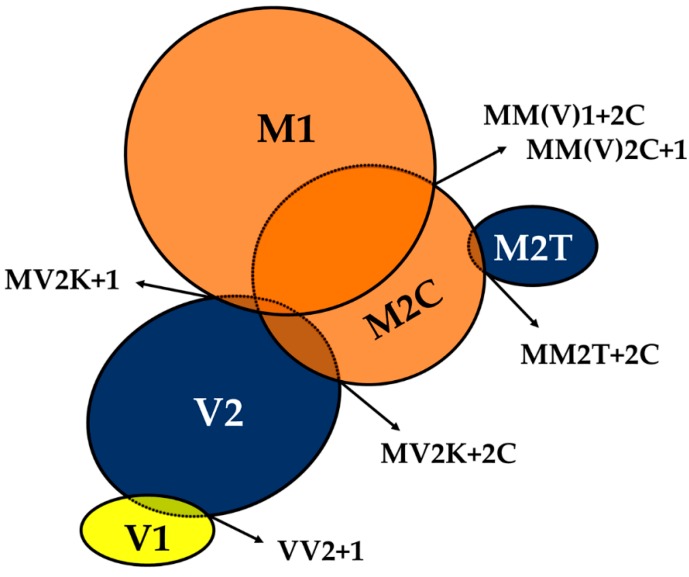
The spectrum of mixed phenotypes and prion strain co-occurrence in sporadic Creutzfeldt–Jakob disease (CJD). Prion strains are classified as “dominant” or “not-dominant” according to their relative quantitative contribution to the mixed CJD phenotype as assessed by histopathological examination and PrP^Sc^ typing. The “dominant-only” strains are highlighted in blue, those either “dominant” or “not-dominant” in orange, and those “not-dominant-only” in yellow. The extent of overlap between circles reflects the relative prevalence of the strain co-occurrence compared to that of the individual “pure” phenotypes. In the M1+M2C strain co-occurrence, M1 dominates over M2C in the majority of cases. For strain nomenclature (e.g. M1, V2) see paragraph 6.

**Figure 3 viruses-11-00309-f003:**
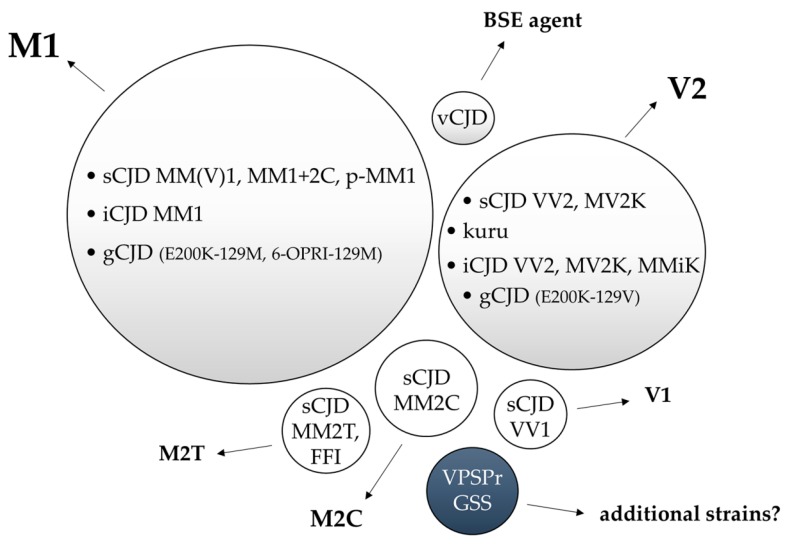
Representation of the relationship between human prion disease subtypes and the isolated strains by experimental transmission. Bubble sizes reflect the relative prevalence of each disease subtype/strain. In light grey, the phenotypes related to prion strains with documented human-to-human transmission. In white, experimentally transmitted phenotypes lacking documented human-to-human transmission. In blue, the prion disorders that transmitted amyloid-PrP in the absence of clinical disease.

**Table 1 viruses-11-00309-t001:** Comparison of clinical, biochemical and transmission data among the sporadic CJD (sCJD) subtypes, variant CJD (vCJD) and variably protease-sensitive prionopathy (VPSPr) (adapted from [68]).

**Human Prion Disorders**	**Phenotype**	**VV2**	**MV2K**	**MM(V)1**	**vCJD**	**MM2T**	**MM2C**	**VV1**	**VPSPr**
	**Frequency** **(% of Cases)**	15	8	40	Rare	<1	<1	<1	~1
	**Mean Duration** **(Months)**	6.3	15.8	4.0	14.0	30.2	20.0	15.3	23.0-45.1
**Biochemical Features**	**T50** **(°C)**	82.05 ± 3.70	79.48 ± 3.63	79.66 ± 2.30	65.26 ± 3.19	59.41 ± 6.04	57.11 ± 5.96	<25	NA
	**PK-Resistance ED50 (U/ml)**	4.14 ± 3.56	1.41 ± 0.89	0.09 ± 0.07	5.19 ± 2.38	0.13 ± 0.09	0.28 ± 0.21	0.03 ± 0.21	0.07 ± 0.01
	**Glycoform Ratio (D/U) ***	2.05	0.88	1.04	2.99	0.66	0.78	0.53	0°
	**Aggregation Ratio ^¥^**	8.68 ± 1.22	8.38 ± 1.93	6.43 ± 2.15	11.65 ± 3.35	10.92 ± 1.44	5.68 ± 0.99	5.47 ± 0.46	3.76 ± 0.28
**Transmission Properties**	**Attack Rate** **(%)**	100	100	100	83–100	93	0	14	0
	**Incubation Time** **(days)**	274 ± 4, 302 ± 9	288 ± 3, 329 ± 3	446 ± 3, 467 ± 24	540 ± 41, 668 ± 22	535 ± 32	NA	568 ± 0	NA
	**Isolated Strain**	V2		M1	BSE	M2T	M2C	V1	ND

^¥^ The aggregation ratio represents a semiquantitative index of the overall PrP^Sc^ aggregation, with values being proportional to the mean size of protein aggregates, as calculated in [67]. PK-resistance ED50 expresses the PK concentration needed to digest 50% of PrP^Sc^. T50 expresses the temperature needed to solubilize 50% of PrP^Sc^. Data are based on densitometric analyses of immunoblots probed with mAb 3F4 or SAF60. Transmission properties refer to those obtained after injection of PrP-humanized knock-in mice expressing normal levels of human PrP^C^ with the most compatible *PRNP* 129 genotype. Attack rate is expressed as the percentage of injected mice that develops clinical signs. Biochemical and transmission properties values are expressed as mean ± standard deviation. * Ratio between diglycosylated and unglycosylated PrP^Sc^ fragment; ° Lacks diglycosylated PrP^Sc^. Data summarized in the Table are taken from the following studies: disease frequency [32,60,69], disease duration [32,60,69,70], T50 [68], PK-resistance ED50, glycosylation and aggregation ratios [67], attack-rate and incubation time [71,72,73,74,75,76,77,78]. NA, not available; ND, not defined.

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
