# Peer review of "Understanding Prion Strains: Evidence from Studies of the Disease Forms Affecting Humans"

_viruses, 2019, doi:10.3390/v11040309_

Reviewer 1 Report

In the literature review, the authors mainly reviewed literature on prion strains of human prion diseases, focusing on pathological features and physicochemical properties of PrPSc and putting emphasis on transmission experiments with relatively physiological conditions in terms of expression levels and expressed PrP, to conclude that currently recognized human prions can be attributable to a limited number of strains. 

The excellent work greatly helps overview the latest knowledge about human prion strains, being very comprehensible and with many references that support their argument.

   Reading the manuscript, I got curious about a couple of points and, adding passages about them might make the manuscript even more informative.  

1) About vCJD prion, Bian et al. propounded that vCJD might actually represent non-adaptive prion amplification (NAPA) of BSE in humans (Jifeng Bian et al., PNAS 2016).

I think that readers who had read the paper would like to know the author’s view on this issue.

2) GSS plaques seem to be associated with the 7-8 kDa fragment, and they can appear independently of spongiform changes and 21-kDa fragment. On the other hand, some human prion diseases show both plaques and spongiform changes without the 7-8 kDa fragments, e.g., vCJD or MV2K.

How different are the constituent PrP molecules between those plaques? What are possible underlying mechanisms for the difference?

I think readers would appreciate those information, if they are in the review.

Minor points

1) In Line 564: Is the “the” before “determining ~” necessary?

2) In Line 473: Is a word missing after the “main” in the sentence “codon 129 genotype is the main responsible for ~”?

Author Response

Response to Reviewer 1 Comments

Point 1: About vCJD prion, Bian et al. propounded that vCJD might actually represent non-adaptive prion amplification (NAPA) of BSE in humans (Jifeng Bian et al., PNAS 2016).

I think that readers who had read the paper would like to know the author’s view on this issue.

Response: We find the concept expressed by Bian and colleagues of interest. Although vCJD behaved quite differently from BSE when inoculated in Tg mice expressing human PrP, which indicated at least a certain degree of adaptation, it is true that vCJD inocula only rarely caused clinical disease. Prion-specific pathology was, instead, observed in the large majority of inoculated animals. We added a brief comment on these data in the revised text (page 15, lines 611-20).  

Point 2: GSS plaques seem to be associated with the 7-8 kDa fragment, and they can appear independently of spongiform changes and 21-kDa fragment. On the other hand, some human prion diseases show both plaques and spongiform changes without the 7-8 kDa fragments, e.g., vCJD or MV2K. 

How different are the constituent PrP molecules between those plaques? What are possible underlying mechanisms for the difference?

I think readers would appreciate that information, if they are in the review.

Response: This point is well-taken. We agree that it is important, especially for the non-specialists, to get the information that in human prion diseases amyloid plaques are not invariably linked to the 7-8 kDa fragments that are ragged at both termini and lack the N-linked glycosylation. We, therefore, revised the text accordingly (page 7, lines 250-54). Regarding the properties of PrP molecules forming plaques in vCJD or MV2K and possible difference in the underlying mechanism of plaque formation between GSS, vCJD and MV2K, there are no definitive explanations, as far as we know.

Minor points

1) In Line 564: Is the “the” before “determining ~” necessary?

2) In Line 473: Is a word missing after the “main” in the sentence “codon 129 genotype is the main responsible for ~”?

Points 1 and 2 have been taken care while editing the article for English.

Reviewer 2 Report

The review by Rossi et al is a thorough, well written account of prion strains and some of their molecular underpinnings. There are, however, a few issues that should be addressed:

In section 4 the authors discuss the PrP fragments generated through PK digestion and raise questions as to the pathogenic role of those fragments (line 238). It is important to note that while the PK generated fragments give information to the structure of the intact protein, these fragments are artificial and may not be generated naturally in vivo. This is in contrast to the N1/C1 and N2/C2 fragments that are naturally occurring.

In section 4 the authors discuss a lot of contrasting data from their own and other groups, such as the amount of protease sensitive PrPsc and the structural stability of the different strains. The authors should discuss why such discrepancies may be occurring and how one should interpret the contrasting data.

Line 100: “physicochemical properties” should be defined.

Line 162: “Pattern A” and “Pattern B” are not indicted in the figure or well defined.

While generally well written the manuscript should be carefully read for typographical errors, for example:

Line 26-27: disease remains not understood

Line 121: MM1 is listed twice instead of MM2

Line 133-134: intermediate molecular between

Line 239: determinism

Line 327: The Soto’s group

Line 473: the main responsible for

Line 534 and 543: Likewise

Line 564: in the determining the host

Line 605-606: allowed to significantly

Line 673: amyloid all protein

Line 685: why some rare

Line 687: why the

Author Response

Response to Reviewer 2 Comments

Point 1: In section 4 the authors discuss the PrP fragments generated through PK digestion and raise questions as to the pathogenic role of those fragments (line 238). It is important to note that while the PK generated fragments give information to the structure of the intact protein, these fragments are artificial and may not be generated naturally in vivo. This is in contrast to the N1/C1 and N2/C2 fragments that are naturally occurring.

Response: I cannot agree on this point.  As discussed in Parchi et al (PNAS 1998), quote “We have shown (Chen et al J Biol Chem 1995) that, in CJD, a significant amount of truncated 21- or 19-kDa PrP-res is formed in vivo. In vivo formation of N-terminal truncated PrP-res fragments also occurs in scrapie-infected mouse brains and scrapie-infected neuroblastoma cells (Caughey et al. J Virol 1991, Borchelt et al, J Cell Biol 1990). Similarly, these as well as previous data indicate that the formation in vivo of PrP-res peptides with ragged N and C termini characterize most, if not all, GSS variants (Tagliavini et al EMBO J 1991, Ghetti et al PNAS 1996). Thus, the in vivo formation of aberrant truncated PrP-res peptides represents a common pathogenetic event in prion disease”.

Point 2: In section 4 the authors discuss a lot of contrasting data from their own and other groups, such as the amount of protease sensitive PrPsc and the structural stability of the different strains. The authors should discuss why such discrepancies may be occurring and how one should interpret the contrasting data.

Response: We extensively discussed the possible causes of such discrepancies in our original studies (Saverioni et al. JBC 2013, Cescatti et al. J Virol 2016). However, to please the reviewer’s request, we have expanded the text and tried to address the potential methodological differences that would at least partially explain the intergroup variability obtained in the characterization of PrPSc properties, and referred the reader to our previous studies for further details (page 9, lines 310-29).

Line 100: “physicochemical properties” should be defined.

Response: We referred the reader to paragrapghs 3 and 4.

Line 162: “Pattern A” and “Pattern B” are not indicated in the figure or well defined.

Response: We better defined the two pattern in the figure.

While generally well written the manuscript should be carefully read for typographical errors, for example:

Line 26-27: disease remains not understood

Line 121: MM1 is listed twice instead of MM2

Line 133-134: intermediate molecular between

Line 239: determinism

Line 327: The Soto’s group

Line 473: the main responsible for

Line 534 and 543: Likewise

Line 564: in the determining the host

Line 605-606: allowed to significantly

Line 673: amyloid all protein

Line 685: why some rare

Line 687: why the

Response: We edited the article for English (see also the title change) and especially corrected all the above-indicated typographical errors. We thank the reviewer for noticing them.

Reviewer 3 Report

Overall this is an excellent and comprehensive review of human prion strains.

A few suggestions/comments:

1) Figure 2 does not contribute to my understanding of the strains and could be deleted.

2) Table 1---needs to be reformatted, it is very difficult to read in its current format.

3) The legend for table 1 needs to be expanded or more information provided in the text. Fro example, it is not clear what incidence refers to (I am assuming percent of cases?). There is little to no information as to how T50 is determined nor what the aggregation ration is. This table summarize the entire manuscript and it would be very helpful if it was easier to read. Perhaps moving the references out of the data part of the table?

4) Could the authors provide possible explanations for the variability in some of the biochemical characterizations between research teams? It has been my experience that, when comparing prion strains, we are often comparing different things---perhaps different disease presentations, different brain regions, the purification of the PrPSc  (or if it was purified), ratio of protease to total protein, primary antibodies used etc. A short section describing some of these potential differences would be helpful for prion researchers and would point out to other readers the underlying difficulties in strain analysis.

5) Line 523--not a complete sentence.

6) LIne 524--properties not proprieties

7) Line 551--Bruce 1994 should be added to the reference list

Author Response

Response to Reviewer 3 Comments

A few suggestions/comments:

Point 1: Figure 2 does not contribute to my understanding of the strains and could be deleted.

Response: We fully understand this comment. Indeed, this figure was not meant to clarify or simplify the issue of the co-occurrence of prion disease subtypes or strains, but rather to show the complexity of the problem, which is one of the declared aims of the review (i.e. to underline the unsolved issues from the human disease perspective). We, therefore, prefer to keep the figure in the review. However, to help the reader, we have implemented figure legend and the text of the specific paragraph (page 6, lines 211-21). 

Point 2: Table 1---needs to be reformatted, it is very difficult to read in its current format.

Response: We have reformatted the table by inverting the orientation (horizontal vs. vertical) of the headings. We also removed the references out of the table and significantly expanded the table legend.

Point 3: The legend for table 1 needs to be expanded or more information provided in the text. For example, it is not clear what incidence refers to (I am assuming percent of cases?). There is little to no information as to how T50 is determined nor what the aggregation ration is. This table summarizes the entire manuscript and it would be very helpful if it was easier to read. Perhaps moving the references out of the data part of the table?

Response: See the reply above (point 2).

Point 4: Could the authors provide possible explanations for the variability in some of the biochemical characterizations between research teams? It has been my experience that, when comparing prion strains, we are often comparing different things---perhaps different disease presentations, different brain regions, the purification of the PrPSc (or if it was purified), the ratio of protease to total protein, primary antibodies used etc. A short section describing some of these potential differences would be helpful for prion researchers and would point out to other readers the underlying difficulties in strain analysis.

Response: As stated before (see response to reviewer#2) we briefly addressed the potential methodological differences that would at least partially explain the intergroup variability obtained in the characterization of PrPSc properties (especially sen-PrPSc), and referred the reader to our previous studies for further details.

5) Line 523--not a complete sentence.

Response: The sentence has been revised.

6) LIne 524--properties not proprieties.

Response: Done

7) Line 551--Bruce 1994 should be added to the reference list.

Response: Done